# Horizontal Transfer and Evolutionary Profiles of Two *Tc1/*DD34E Transposons (*ZB* and *SB*) in Vertebrates

**DOI:** 10.3390/genes13122239

**Published:** 2022-11-29

**Authors:** Wenzhu Jia, Emmanuel Asare, Tao Liu, Pingjing Zhang, Yali Wang, Saisai Wang, Dan Shen, Csaba Miskey, Bo Gao, Zoltán Ivics, Qijun Qian, Chengyi Song

**Affiliations:** 1College of Animal Science and Technology, Yangzhou University, Yangzhou 225009, China; 2Shanghai Cell Therapy Group, Shanghai 201805, China; 3Division of Medical Biotechnology, Paul Ehrlich Institute, 63225 Langen, Germany

**Keywords:** horizontal transfer, transposon, *SB*, *Tc1/mariner*, *Tc1/*DD34E, *ZB*

## Abstract

Both ZeBrafish (*ZB*), a recently identified DNA transposon in the zebrafish genome, and *SB*, a reconstructed transposon originally discovered in several fish species, are known to exhibit high transposition activity in vertebrate cells. Although a similar structural organization was observed for *ZB* and *SB* transposons, the evolutionary profiles of their homologs in various species remain unknown. In the present study, we compared their taxonomic ranges, structural arrangements, sequence identities, evolution dynamics, and horizontal transfer occurrences in vertebrates. In total, 629 *ZB* and 366 *SB* homologs were obtained and classified into four distinct clades, named *ZB*, *ZB*-like, *SB,* and *SB*-like. They displayed narrow taxonomic distributions in eukaryotes, and were mostly found in vertebrates, Actinopterygii in particular tended to be the major reservoir hosts of these transposons. Similar structural features and high sequence identities were observed for transposons and transposase, notably homologous to the *SB* and *ZB* elements. The genomic sequences that flank the *ZB* and *SB* transposons in the genomes revealed highly conserved integration profiles with strong preferential integration into AT repeats. Both *SB* and *ZB* transposons experienced horizontal transfer (HT) events, which were most common in Actinopterygii. Our current study helps to increase our understanding of the evolutionary properties and histories of *SB* and *ZB* transposon families in animals.

## 1. Introduction

Class II transposable elements are DNA segments (jumping genes) that can mobilize and integrate into the genome by mechanisms involving a DNA intermediate [1]. Transposons can form a substantial fraction of vertebrate genomes (4–60%) [2], and can have considerable impact on genome function based on their ability to move and reorganize the DNA. Most DNA transposons can be classified into families of autonomous (encoding a functional transposase) and nonautonomous (lacking a functional transposase) elements, characterized by their ability to respond (be mobilized) to the same transposase. Transposons belonging to the same family typically share several nucleotides in their termini that are identical [3]. Similarly, the superfamilies can also be identified by amino acid analysis sequence of the transposase genes, both in eukaryote and prokaryote transposons [4].

The transposition mechanism of a widespread [5] and extensively characterized “cut and paste” transposon superfamily, *Tc1*/*mariner,* is based on the excision (cut) and reinsertion (paste) of fragments into a new location in the genome. As an outcome of transposon integration, the element generates a target site duplication [6]. *Tc1/mariner* superfamily transposons are about 1.6 kb in length and are characterized by terminal inverted repeats (TIRs) 17–300 bp in length that flank a coding sequence for a transposase of around 340 amino acids [7] which catalyzes the transposition reaction [8,9]. Subgroups of *Tc1*/*mariner* elements have long TIRs (type 1 TIRs) with internal direct repeats. The superfamily of *Tc1*/*mariner* shares an amino acid sequence motif in the catalytic domain denoted with the DDE(D) triad which is an integral part of the catalytic site [8]. The literature has reported a high level of internal diversity in the *Tc1/mariner* DNA transposons. There are ten discovered families of *Tc1/mariner*:DD34E/*Tc1*, DD35E/*TR*, DD36E/*IC*, DD37E/*TRT*, DD38E/*IT*, DD34D/*mariner*, DD37D/*maT*, DD39D/*GT*, DD41D/*VS* [10].Notably, the intra-family structure of DD34E/*Tc1* transposons is very complex and still poorly understood. However, DD35E/*TR* [11], DD36E/*IC* [12], DD37E/*TRT* [13], and DD38E/*IT* [14] seem to belong to intra-families of DD34E/*Tc1* based on phylogenetic analysis.

A highly active DNA transposon, ZeBrafish (*ZB*), which was named after its source exhibits high transposition activity in vertebrate cells, in the range of those of the most widely used transposons *piggyBac* (*PB*) and *Sleeping Beauty* (*SB*) [10,15]. Notably, *ZB* has displayed a distinct potential as a gene transfer tool for transgenesis and mutagenesis in animals and human cells [10]. *SB* is an engineered transposon that was rebuilt based on DNA transposon fossils from salmonid fish genomes [9]. The transposition activity of the first version of *SB* (*SB*10) was relatively low, but a hyperactive mutant was obtained after several rounds of optimizations, resulting in an updated version of the *SB* transposase (*SB*100X) that displays a transpositional efficiency about 100 times higher than the original *SB* transposase [16,17]. The *SB* transposon was extensively applied as a molecular tool for transgenesis, gene therapy, and functional genomics (such as gene trapping, enhancer trapping, and cancer gene trapping). *SB* was considered for gene therapy ex vivo and in vivo [18]. In addition, extensive reports indicate substantial progress was recently achieved towards developing *SB* transposon-based chimeric antigen receptor (CAR) T cell engineering technology, which holds great promise for the future treatment of cancer [19]. Indeed, clinical trials with *SB*-engineered cells are currently running [20].

Comparatively, similar structural organization and target site sequence preferences were observed for *ZB* and *SB*, but *ZB* had a slightly different integration profile as compared with the features of *SB* at the mammalian genome-wide scale. Namely, *ZB* displays preferential integration towards transcriptional regulatory regions of genes, whereas *SB* integrates into the genome in an almost completely random fashion [10]. As a consequence, *ZB* was mainly applied for enhancer trapping in zebrafish, sperm mutagenesis in mice, and increasing the integration rate in the germline genome in transgenic chickens [10,15]. Both *ZB* and *SB* transposases contain the conserved catalytic amino acid triad motif (DD34E) and were classified as DD34E/*Tc1* transposons according to the phylogenetic tree analysis by Shen et al. [10].

The evolution profiles, including taxonomic distribution, evolution dynamics, and evolution patterns, of *ZB* and *SB* transposons remain largely unknown. Gaining further insight into the evolution landscape of DNA transposons will provide a better understanding of their impact on genome evolution and application. Here, we systematically investigated the taxonomic distributions, structural organizations, sequence identities, and evolution dynamics, and we also provide evidence to support the occurrence of horizontal transfers (HTs) of *ZB* and *SB* across vertebrates.

## 2. Materials and Methods

### 2.1. Transposon Searching

The taxonomic distribution of *ZB* and *SB* transposons was determined via TBLASTN (v. 2.12.0) [21] searching in the National Center for Biotechnology Information (NCBI) against the accessible assembled eukaryote genomes (including Contig and Scaffold). This entailed using the queries of the full-length transposase protein sequences of 340 aa *SB100X* [22] and 341 aa *ZB* [10], with defaulted algorithm parameters and an E-value of 1 × 10^−4^. The obtained transposases were used as queries to identify other homology elements in succession. Finally, all mined transposases were submitted for phylogenetic analysis and only sequences belonging to *ZB* and *SB* clades were used for further analysis. The best hits of *ZB* and *SB* elements (E-value of 1 × 10^−4^) were extracted with 2 kb flanking sequences in each genome, and their boundaries in each genome were defined by alignment using the BioEdit tool (v. 7.2.0) in the ClustalW program [23] and were observed manually for any changes. The representative sequence (<10 copies, hard to derive consensus sequence) or consensus sequence (>10 copies) in each genome was submitted to BLASTN for each host genome to estimate copy numbers. More than 40% coverage and 90% identity of BLAST hits to the queries (with a default E value) were used to calculate copy numbers for each element to avoid overlapping hits between subfamilies.

### 2.2. Phylogenetic Analysis

The coding sequences (CDS) of the identified *SB* and *ZB* elements were aligned with the CDS sequences of 29 known DNA transposases representing seven families (DD34E*/Tc1*, DD36E*/IC*, DD37E*/TRT*, DD41D*/VS*, DD39D/*GT*, DD34D*/mariner*, and DD37D*/maT*) of the *Tc1/mariner* transposon family by MAFFT v. 7.310 [24]. Then, the alignment was submitted to the IQ-tree program (v. 1.6.12) [25] to determine their evolutionary relationships by using the maximum likelihood method, with an ultrafast bootstrap value of 1000. The DD37D, which forms a distinct clade with DD41D and DD39D from DD34E*/Tc1* [12], was used as the outgroup. The best-suited amino acid substitution model was selected by ModelFinder embedded in the IQ-tree program (v. 1.6.12) [26].

### 2.3. Sequence Analysis

The TIR was determined manually using the ClustalW program in the BioEdit tool (v. 7.2.0) [23]. The sequences of TIR, CDS, DBD(DNA-binding domain), and DDE of *SB* and *ZB* elements or transposases were aligned using MAFFT v. 7.310, while the sequence identities were calculated using the BioEdit tool (v. 7.2.0) [23]. When calculating the sequence identity of TIRs, the BioEdit tool filtered out sequences that were too short, incomplete, or had only one TIR. The obtained sequence identities were drawn with the HeatMap program in the TBtools (v. 1.0987663) (https://github.com/CJ-Chen/TBtools/releases, accessed on 12 March 2022) [27]. Nuclear localization signal (NLS) motifs were predicted using PSORT (https://wolfpsort.hgc.jp/, accessed on 13 March 2022). Helix prediction of the searched transposase was performed using PSIPRED (v3.2) (http://bioinf.cs.ucl.ac.uk/psipred/, accessed on 15 March 2022) [28]. Multiple alignments were performed using the multiple alignment program ClustalW of the BioEdit tool (v. 7.2.0) and manually edited and annotated using GeneDoc (v. 2.7.0.0) [29]. The structure predicted by the transposase was presented by the tool Illustrator for Biological Sequences (IBS v. 1.0.3) [30]. In addition, the sequence logos were created by Weblogo v. 3.7 (http://weblogo.threeplusone.com/create.cgi, accessed on 16 March 2022). The default parameters were used in the programs of this subsection.

### 2.4. HT Analysis

The pairwise distances between the host genes and the transposons were used to detect *ZB* and *SB* transposons’ horizontal transfer events. Two globally conserved ribosomal proteins (RPL3 and RPL4), were selected as the host genes and successfully applied to evaluate the HT events for *hAT* and *Tigger* transposons [31,32]. The pairwise distances between transposase-coding sequences (*ZB* and *SB*) and host gene-coding (RPL3 and RPL4) sequences were calculated to detect possible HT events of transposons. Transposons with a sequence identity of less than 70% of pairwise species were excluded from the HT analysis. To decrease the putatively false-positive estimation of HT events, the HT events were recognized when the genetic distance of transposons between species was 1.2 times smaller than both the host genes (RPL3 and RPL4).

All accessible gene annotations (CDS) for host genes (RPL3 and RPL4) of species involved in the putative HT events of *ZB* and *SB* were retrieved from the NCBI database. The CDS of these genes were searched against the NCBI genome database via TBLASTN for those species whose host genes were not annotated and manually annotated by GenScan (http://hollywood.mit.edu/GENSCAN.html, accessed on 16 March 2022). The multiple sequence alignments of the host-gene- CDS and transposase- CDS was built by the MAFFT program (v. 7.310) [25] and subsequently submitted to MEGA software (v. 7.0.26) to calculate the genetic distances between the host genes and transposons (pairwise deletion and maximum composite likelihood) [33]. The genetic distances between host genes and transposons in each species are listed in Appendix A, and the alignment files were deposited as Appendix A. It was summarized using GraphPad Prism v. 8.0.1.244.

## 3. Results

### 3.1. Phylogeny and Sequence Analysis of ZB and SB Transposons

Overall, 629 sequences homologous to *ZB* and 366 sequences homologous to *SB* were obtained and submitted for phylogenetic analysis. The phylogenetic tree showed that all identified *SB* and *ZB* homology elements belonged to the clade of *DD34E/Tc1*, and they formed four distinct branches with strong bootstrap supports (100). The branches harboring 341 aa *ZB* and 340 aa *SB*100X reference sequences were named *ZB* and *SB*, respectively, and their close sibling branches were named *ZB*-like and *SB*-like, respectively (Figure 1 and Appendix A). Overall, *ZB* elements from 313 species, *ZB*-like elements from 316 species, *SB* elements from 108 species, and *SB*-like elements from 258 species were designated as *ZB*, *ZB*-like, *SB*, and *SB*-like transposons, respectively (Appendix A).

Pairwise sequence comparison of *SB* and *ZB* transposons revealed that the CDS sequences (DNA sequences) of *ZB*, *ZB*-like, *SB*, and *SB*-like transposases are highly conserved. Their internal sequence identities were higher than 80%, with *ZB* and *SB*-like represented by over 90%, and *ZB*-like and *SB* by over 80% (Figure 2A). In contrast, the sequence identities between *ZB* and *ZB*-like, and between *SB* and *SB*-like were 61%. The sequence identities of CDS between *SB* and *ZB* groups ranged from 52% to 54% (Figure 2A). Furthermore, the DDE domains (protein sequences) tended to be more conserved than the DBD (protein sequences) domains. The sequence identities of DDE domains between *ZB* and *ZB*-like and between *SB* and *SB*-like were 68%. In comparison, the sequence identities of DDE between *SB* and *ZB* groups ranged from 58% to 61% (Figure 2B,C). Similarly, the sequence identities of DBD domains between *ZB* and *ZB*-like, and between *SB* and *SB*-like, were 54% and 55%, respectively. Whereas, the sequence identities of DBD between *SB* and *ZB* groups range from 34% to 38% (Figure 2B). In addition, their internal sequence identities of TIRs for *ZB*, *ZB*-like, *SB,* and *SB*-like are 78%, 68%, 65%, and 80%, respectively. The sequence identities of TIRs between *ZB* and *ZB*-like, and between *SB* and *SB-lik,e* were 39% and 33%, respectively. In contrast, the sequence identities of TIRs between *SB* and *ZB* groups range from 20% to 30% (Figure 2D).

### 3.2. Taxonomic Distribution and Phylogenetic Analysis of ZB and SB Transposons

The *ZB* and *SB* homology transposons display narrow taxonomic distributions in eukaryotes. They were only detected in animals, mostly in vertebrates but also a few lineages were found in invertebrates. Indeed, different taxonomic distributions of four branches (*ZB*, *ZB*-like, *SB*, and *SB*-like) were observed (Figure 3A and Table 1). *ZB* was observed in 299 Actinopterygii species (32 orders), 9 Anura species of vertebrates, 4 Arthropoda species of invertebrates, and detected in only 1 Mollusca species (*Euprymna scolopes*). Although *ZB*-like is more widely distributed in vertebrates than in other elements, it was discovered in 271 Actinopterygii species (42 orders), 17 Anura species, 5 Agnatha species (2 orders), 19 Squamata species, 1 Sarcopterygii species, 2 Aves species, and 1 Chondrichthyes species. *SB* was mostly detected in Actinopterygii of vertebrates (107 species of 33 orders) and has only been seen in one Mollusca (*Anentome helena*) species. Whereas, *SB*-like invaded into 255 Actinopterygii species (25 orders), 1 Aves species of vertebrates, 1 Echinodermata species (*Lytechinus variegatus*) and 1 Cnidaria (*Dendronephthya gigantea*) of invertebrates (Figure 3A and Table 1).

Furthermore, Actinopterygii is the major host of *SB* and *ZB* transposons, with 107 species for *SB*, 255 species for *SB*-like, 299 species for *ZB*, and 271 species for *ZB*-like detected in this lineage, respectively (Figure 3B). However, substantially different distribution patterns in the orders of Actinopterygii were observed for the four transposon branches. The *SB*-like and *ZB* were widely distributed in the order of Cichliformes, with 202 species and 193 species detected, accounting for 78% and 62% of the total detected species, respectively.Whereas, *ZB*-like and *SB* are more evenly distributed in the orders of Actinopterygii (Figure 3B). In addition, we found that some branches of *SB*, *SB*-like*, ZB*, and *ZB*-like co-exist in some species. A total of 2 species (*Siniperca knerii* and *Siniperca scherzeri*) are co-invaded by all four branches, 3 to 24 species are co-invaded by three branches, and 16 to 159 species are co-invaded by two branches (Figure 3C and Appendix A).

### 3.3. Structural Organization of ZB and SB

Generally, similar structural organization was observed for *ZB* and *SB* transposons. The total lengths of intact *SB* and *ZB* transposons range from 1.3 kb to 3.0 kb, but most of them (96% of the total detected intact transposons, 336/350) are between 1.5 kb and 1.7 kb. They contain a single ORF (open reading frame) that encodes a transposase of about 340 aa, ranging from 302 aa to 411 aa, and flanked with TIRs in lengths varied from 27 bp to 415 bp (Table 1, Figure 4A, and Appendix A). Overall, the structural features of *SB* and *ZB* transposons are similar to that observed for other *Tc1/mariner* members [10,14,35].

The CDS of *ZB*, *ZB*-like, *SB,* and *SB*-like transposases are highly conserved, and display over 80% of sequence identities (Figure 2A). In the intact *SB* and *ZB* transposases, several well-defined domains, including the catalytic domain, GRPR-like motif, linker motif [K(V/T)PLLS], nuclear localization sequence (NLS), and DNA-binding domain (DBD), which contains six helixes in the N terminus, and were identified and indicated in Figure 4A,B. The interdomain linker (Figure 4B and Appendix A) was identified as a conserved sequence stretch (KKPLLS) in *SB100X* transposase [36]. The first and last two residues of linkers varied across *ZB*, *ZB*-like, *SB,* and *SB*-like transposons, while the middle three residues (PLL) are highly conserved (Appendix A).

Compared with other *Tc1/mariner* families, such as *maT/*DD39D*, GT/*DD37D, and *IC/*DD36E [12,13], most TIRs of *SB* and *ZB* elements (94% of the total detected intact transposons, 330/350) are relatively long, ranging from 180 bp to 300 bp. However, very long and short TIRs are also observed in some species (Table 1 and Appendix A). The 3′ TIR of *SB* partially overlaps with the ORF regions, which is not observed for the other three group transposons (Figure 4C). The end motifs (20 bp) of TIRs are highly conserved across *ZB*, *ZB*-like, *SB*, and *SB*-like, and start with a GC-rich motif followed by an AT-rich region (Figure 5A). The genomic flank sequence analysis revealed that the integration profiles of *ZB*, *ZB*-like, *SB*, and *SB*-like in genomes are highly conserved, and display strongly preferential integration into AT repeats (Figure 5B).

### 3.4. Evolution Dynamics of ZB and SB Transposons

The genomic copy numbers of *ZB*, *ZB*-like, *SB,* and *SB*-like transposons vary significantly across species, ranging from one to several thousand (Appendix A). Overall, 142 (45.37%, 142/313), 205 (64.87%, 205/316), 54 (50%, 54/108), 63 (24.42%, 63/258) genomes harbor complete copies (transposons flanked by detectable TSDs and TIRs) of *ZB*, *ZB*-like, *SB*, and *SB*-like transposons, respectively. Most high numbers of full copies (>=100) of *ZB*, *ZB*-like, *SB*, and *SB*-like were detected in Actinopterygii. However, high numbers (>=100) of full copies of *ZB*-like were also detected in Agnatha, Sarcopterygii, Anura, Squamata, and Chondrichthyes (Table 2). Furthermore, intact copies (transposons flanked by detectable TSDs and TIRs and encoded >=300 aa transposases) were detected in many species of multiple animal lineages for all four groups of transposons, but with significant variations across groups and lineages, which support that these transposons display recent and current activities in some lineages of animals, but with differential evolution dynamics (Table 2). In general, 105 (33.55%, 105/313), 183 (57.91%, 183/316), 26 (24.07%, 26/108), and 36 (13.95%, 36/258) genomes contain an intact copy of *ZB*, *ZB*-like, *SB*, and *SB*-like transposons, respectively. However, most of them represent less than 10 intact copies in genomes, and only 4, 3, 57, and 51 species contain 10 to 99 intact copies of *SB*, *SB*-like*, ZB*, and *ZB*-like transposons in their genomes, respectively. Very high intact copies (>=100) were observed in very few species (one or four) for *SB* and *SB*-like, and not detected for *ZB*, but observed for many species (26) for *ZB*-like (Table 2).

Overall, *ZB*-like has been significantly amplified in some genomes of animals (more than 100 copies), and the intact copy number of *ZB*-like is much higher than that of *ZB* in most species, indicating *ZB*-like may be more active than *ZB* in most lineages of animals. More than 100 *ZB*-like intact copies were detected in 26 species (1 Sarcopterygii species, 19 Actinopterygii species, 3 Agnatha species, and 3 Anura species). The most significant number of intact *ZB*-like copies (5188) was detected in *Microcaecilia unicolor* (Sacopterygii). At the same time, *ZB* represents the highest intact copy number in *Salarias fasciatus* of Actinopterygii by only 81. Furthermore, except for *Parhyale hawaiensis* in Arthropoda, all species with more than ten intact copies of *ZB* are distributed in Actinopterygii, and 80% of these belong to order Cichliformes (Table 2 and Appendix A).

While *SB* and *SB*-like have undergone significant expansion in some Actinopterygii species (more than 100 copies), the numbers of intact *SB*-like copies in some genomes are higher than that of *SB*, suggesting that *SB*-like transposons tend to be more active than *SB*. The species containing intact copies of *SB* and *SB*-like species are much less than that of *ZB* and *ZB*-like, and only several species in Actinopterygii harbor 10–99 or more than 100 intact copies of *SB* and *SB*-like in their genomes (Table 2 and Appendix A). In addition, *SB* transposons were detected in 13 species of Salmonid, where the original *SB* transposase was reconstructed based on the inactive copies from multiple species [9], but *SB* elements in most species tend to be truncated in Salmonid genomes, which agree with previous studies [9]. However, more than 100 intact copies of *SB* in *Coregonus clupeaformis* of Salmonid were detected, indicating that *SB* may still be active in some species of Salmonid (Appendix A).

### 3.5. Most ZB and SB Transposons Obtained by Horizontal Transfer

The HT events of *ZB* and *SB* transposons were recognized based on the standards described in methods and summarized in Appendix A. The number of species involved in HT events was illustrated in Figure 6, indicating that HT obtained most *ZB* and *SB* transposons in animals. Overall, 252 (80.5% of the total detected species) *ZB*, 184 (58.2% of the total detected species) *ZB*-like, 71 (65.7% of the total detected species) *SB*, and 241 (93.4% of the total detected species) *SB*-like invaded species were involved in HT events (Figure 6A). Moreover, most HT events were confirmed in Actinopterygii at which the recorded species involved in HT events were *SB*: 64.4%, *ZB*: 84.3%, *ZB*-like: 66.1%, of which the highest occurred in *SB*-like: 94.5%. Notaby, most were detected in Cichliformes and Perciformes (Figure 6C,D). However, some HT events were observed in Squamata, where 26.3% (5/19) *ZB*-like invaded species are involved in HT events (Figure 6D). In addition, 16 species of Actinopterygii tend to be more common for HTs of these transposons and have been invaded by at least three families. Particularly, co-HT events of the four families (*ZB*, *ZB*-like, *SB*, and *SB*-like) were detected for three species (*Siniperca knerii*, *Siniperca scherzeri*, and *Mastacembelus armatus*) and represent the most common species of HTs (Figure 6B).

## 4. Discussion

### 4.1. Recent Origins of ZB and SB Transposons

The phylogenetic relationships of IS630-*Tc1-Mariner* (*ITm*) transposons were recently reviewed and at least four superfamilies were suggested, including DDxD/*pogo*, DD34E/*Gambol*, *Tc1/mariner*, and DD82E/*Sailor*. DD82E/*Sailor* is a new superfamily characterized recently, with a DD82E catalytic domain distinct from the other three groups (DD34E/*Gambol*, and *Tc1/mariner*) [5,37,38,39]. Both DD34E/*Gambol* and DD82E/*Sailor* superfamilies seem to represent low diversity and narrow distribution in nature, while extremely high diversity and wide distribution were observed for DDxD/*pogo* and *Tc1/mariner* superfamilies. Six distinct families (*Passer*, *Tigger*, *pogo*R, *Lemi*, *Mover*, and *Fot/Fot*-like) were detected for DDxD/*pogo* transposons [38], while at least nine distinct families (DD34E/*Tc1*, DD35E/*TR*, DD36E/*IC*, DD37E/*TRT*, DD38E/*IT*, DD34D/*mariner*, DD37D/*maT*, DD39D/*GT*, and DD41D/*VS*) have been defined for *Tc1/mariner* transposons [14]. Furthermore, a previous study from Gao et al. [40] also demonstrated that DD34E/*Tc1* transposons display a high diversity at the family level because at least five distinct clusters or sub-families (*Passport*-like, *SB*-like, *Frog Prince*-like, *Minos*-like, and *Bari*-like) were identified. DD34E/*Tc1* transposons exhibit an unexpected diversity and may evolve into many families as a common ancestor. It was recently indicated that at least three families (DD35E/*TR*, DD36E/*IC*, and DD38E/*IT*) displaying the closest phylogenetic relationship and highest sequence identity to DD34E/*Tc1* transposons may have evolved from this family. Here, we systematically defined the evolution profiles of *ZB*, a naturally active transposon from zebrafish [10], and *SB*, a rebuilt active transposon [41], which belong to DD34E/*Tc1* transposons. Overall, four distinct clades named *ZB*, *ZB*-like, *SB*, and *SB*-like were identified and exhibited the closest phylogenetic relationship with the DD34E/*Tc1* family with typical structure organizations of this family. Generally, *ZB*, *ZB*-like, *SB*, and *SB*-like displayed a similar evolution profile and share a high sequence identity. The *ZB*, *ZB*-like, *SB*, and *SB*-like displayed a narrow taxonomic distribution, and are mainly detected in vertebrates (particularly in Actinopterygii), which is similar to that of DD35E/*TR*, DD36E/*IC*, and DD38E/*IT*, and different from that of DD37D/*maT*, DD39D/*GT*, and DD41D/*VS*. DD37D/*maT* and DD41D/*VS* mainly distribute in invertebrates, while DD39D/*GT* in plants.

Additionally, our data analysis also revealed that *ZB*, *ZB*-like, *SB*, and *SB*-like displayed high intra-family and inter-family sequence identities, and intact copies were detected in many species of multiple animal lineages for all four groups of transposons. The intact copy number and sequence identity of a transposon in a given genome are key factors to judge the activity in the genome. A high number of intact copies means the transposons have obtained a substantial amplification and may still be active, and indeed that they can jump in the genome. More accurate predictions of activity can be obtained through the combination of more data analysis including structure organization and K divergence. Our data analysis indicates that *ZB*, *ZB*-like, *SB*, and *SB*-like are recently evolved families, and represent recent and current activity. Furthermore, it indicates that more active members may exist in diverse species of animals, beside *ZB*, which was proven as a highly active element in zebrafish [10]. However, their transposition activities need further experimental validation.

### 4.2. Horizontal Transfer of ZB and SB Transposons

The horizontal transfer (HT) has long been recognized as an important driver for species diversity and has evolutionary significance on the nuclear genomes within prokaryote domains (bacteria and archaea) [31,39,42]. It was once believed that HT in eukaryotes are rare; yet shreds of evidence support that HT events of mobile elements, including DNA transposons and retrotransposons, are common in eukaryotes and may contribute to shaping genomic and evolutionary patterns in eukaryotes [43,44,45,46]. Although the mechanisms of HTs are still largely unknown, the close physical relationship between a parasite and its host could facilitate horizontal transfer [47,48,49].

The HT events of retrotransposons between kingdoms of eukaryotes (from Arthropods to Flowering Plants) or between phyla were observed [13,44,45,46]. DNA transposons are widespread across eukaryote kingdoms. The HT events of *hAT* DNA transposons were observed in multiple lineages of animals, and they may play a role in shaping the evolution of animal genomes [31,48,50]. While *Tc1/Mariner* superfamily appears to be the most common type of TEs among other DNA transposons involved in HT [51]. Most well-defined families of *Tc1/mariner* families, including DD35E/*TR* [11], DD36E/*IC* [12], DD37E/*TRT* [52], DD38E/*IT* [14], DD37D/*maT* [13], DD39D/*GT* [13], and DD41D/*VS* [53], are involved in HT events [50]. In this study, our data analysis indicated that *ZB* and *SB* transposons in animals are largely obtained by HT events, mainly occurring in Actinopterygii, which were also observed for DD35E/*TR* [11], DD36E/*IC* [12], DD37E/*TRT* [13], DD38E/*IT* [14], and DD41D/*VS* [53]. This indicated that Actinopterygii tend to be “hot” hosts of HTs of *Tc1/mariner* transposons. On the other hand, high diversity and common HT events of *Tc1/mariner* transposons in Actinopterygii also suggest that this superfamily may play roles in shaping the evolution of genomes and contribute to the speciation of this lineage. A similar biological role was observed for the *Tigger* transposons, a family of *pogo* transposons. These transposons (*Tigger*) are commonly involved in HT events across different lineages of animals, including mammals, that may have contributed to their wide taxonomic distribution, indicating that *Tigger* may play a role in the shaping of mammal genome evolution [32].

### 4.3. Structure Organization of ZB and SB Transposons

In the present study, *SB* and *ZB* are elements of the DD34E/*Tc1* group, which present the typical structural organization of *Tc1/mariner* elements [5,8]. Functional domain analysis indicated that both *ZB* and *SB* transposases have distinct transposition active domains, including a DNA binding domain, catalytic domain (DDE), nuclear localization signal (NLS), GRPR-like motif, and a linker motif. The linker [K(V/T)PLLS] is suggested to be structurally equivalent to the regulatory WVPHEL motif of *mariner* transposases [36], and was shown to play a critical role in orchestrating cleavage events within the transpososome [35,54,55]. According to the present study, most TIRs of *SB* and *ZB* elements are relatively long, ranging from 180 bp to 300 bp. However, very long or short TIRs are also observed in some species. The 3′ TIR of *SB* partially overlaps with the ORF regions, which is not observed for the other three group transposons. The end motifs (20 bp) of TIRs are highly conserved across *ZB*, *ZB*-like, *SB*, and *SB*-like and start with a GC-rich motif followed by an AT-rich region. Analyses of genomic flanking sequences revealed that *ZB*, *ZB*-like, *SB*, and *SB*-like integration profiles in genomes are evolutionarily conserved and show a distinct preference for integration into AT repeats. *ZB* had a longer TIR sequence (TIR:201 bp) and preferred to integrate into the regions containing long repeated dinucleotide TA sequences similar to *SB* [56,57]. Similarly, the present study shows that the general structure organization of *ZB* and *SB* is similar to two close families of *Tc1/*DD34E superfamily transposons. *ZB* was identified to share a similar structural organization and target site sequence preference but there exists a slightly different integration profile compared with the features of *SB* at the mammalian genome-wide scale [10]. In the *Tc1/mariner* family, the inverted repeats vary in length and contain transposase-binding sites in different numbers and patterns, thus based on the distinct “DDE/D” signatures of transposase. Furthermore, the DDE domains (protein sequences) tend to be more conserved than the DBD domains, and the sequence identities of DDE domains are 68% between both *ZB* and *ZB*-like*,* and *SB* and *SB*-like Comparatively, *ZB* and *SB* are derived from a common ancestor of DD34E/*Tc1*. *ZB* shares the same clade with DD34E/*Passport*, while *SB* also shares the same clade with DD34E/*Quetzal* [56].

## 5. Conclusions

In the present study, we established that four distinct clades of transposons (*ZB*, *ZB*-like, *SB*, and *SB*-like) exhibited the closest phylogenetic relationship with typical structure organizations to the DD34E/*Tc1* family. In addition, *SB* and *ZB* displayed a narrow distribution in eukaryotes but were mostly detected in animals. Similarly, evidence to support the occurrence of HT events of *ZB* and *SB* across vertebrates indicated that these transposons largely occurred in animals, and specifically mainly in Actinopterygii. The current study provides a further understanding of the evolutionary history of *ZB*, *SB*, and *Tc1/mariner* elements, and updates the classification of DD34E/*Tc1*.

## Figures and Tables

**Figure 1 genes-13-02239-f001:**
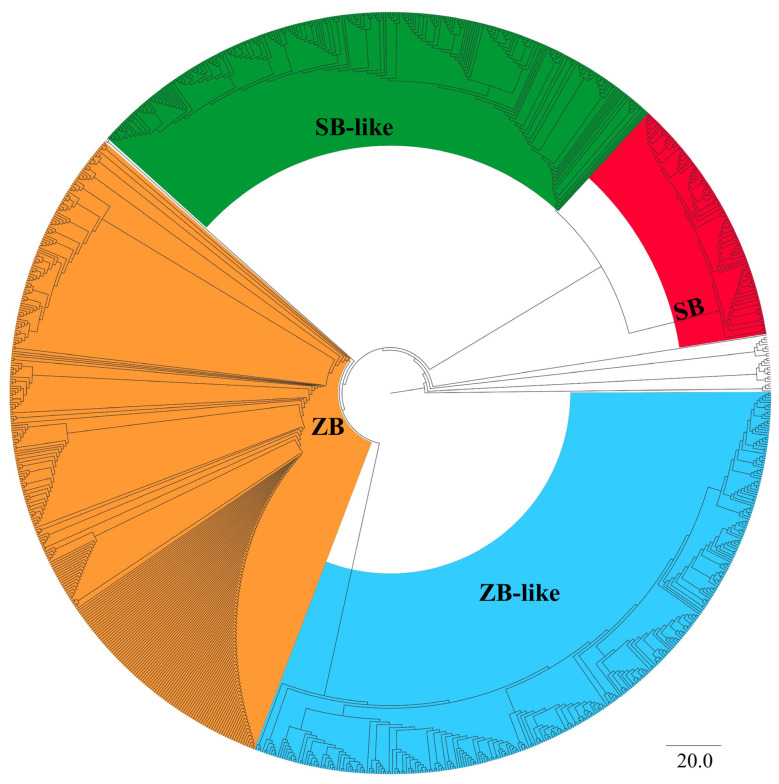
Phylogenetic analysis of *SB* and *ZB* elements. Bootstrapped (1000 replicates) phylogenetic trees were inferred using the maximum likelihood method in IQ-TREE (v. 1.6.12) [34]. Reference families were DD34E*/Tc1*, DD36E*/IC*, DD37E*/TRT*, DD41D*/VS*, DD39D, and DD34D*/mariner*, DD37D*/maT*. DD37D*/maT* was used here as an outgroup. *SB* and *ZB* elements form a separate branch individually. The red and green parts in the figure represent the two branches that make-up *SB* elements, namely cluster *SB* and cluster *SB*-like, and the orange and blue parts in the figure represent the two branches that make-up *ZB* elements, namely cluster *ZB* and cluster *ZB*-like, with a confidence level of 100. See Appendix A for details.

**Figure 2 genes-13-02239-f002:**
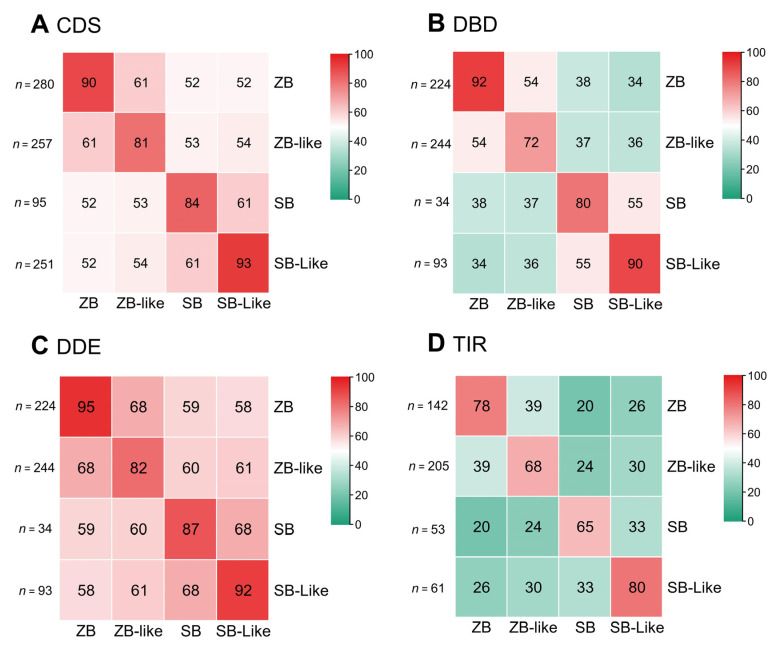
Sequence identities between *ZB* and *SB* elements among species. Sequence identities were measured by pairwise comparison of representative sequences of *SB* and *ZB* elements. The numbers in the heatmap are the percentage of the average values of the sequences’ identities of the two types of transposons in the corresponding row and column, and “*n*” represents the number of each type of transposon sequence (**A**–**D**). The average values of sequence identities were measured by pairwise comparison of representative sequences of CDS (**A**), DBD (**B**), DDE (**C**), and TIR (**D**) of *ZB* and *SB* elements.

**Figure 3 genes-13-02239-f003:**
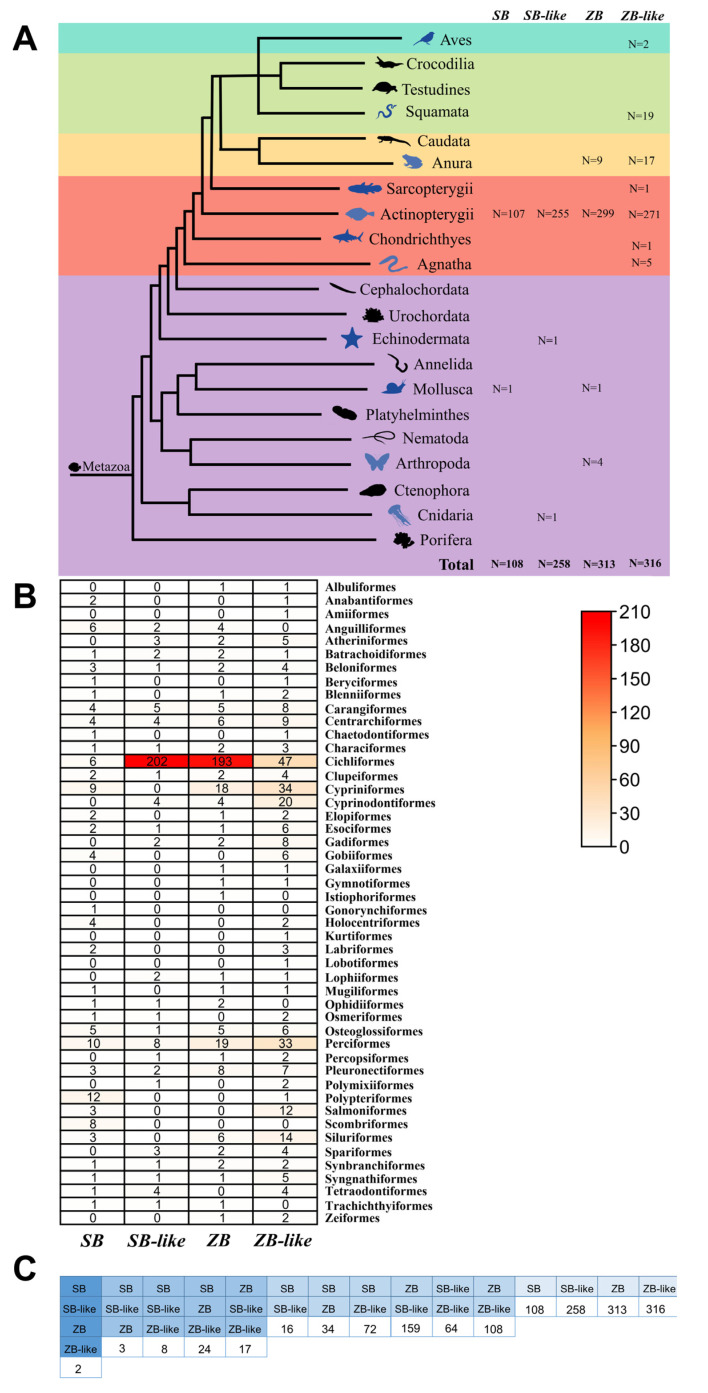
Taxonomic distribution of *SB* and *ZB* elements. (**A**) Taxonomic distribution of *SB* and *ZB* elements in the Metazoa, including cluster *SB*, cluster *SB*-like, cluster *ZB,* and cluster *ZB*-like. The numbers following the animal panels represent the number of species for which the corresponding transposon was detected in that lineage. (**B**) Taxonomic distribution of *SB* and *ZB* elements in the Actinopterygii. This graph was drawn based on statistics (Appendix A). The numbers in the table represent the distribution of *SB* and *ZB* species in this order. The darker the color, the more species containing the corresponding transposon in that order. (**C**) Overlapping species of *SB* and *ZB*. The numbers in the figure represent the number of species that possess several types of transposons in this column.

**Figure 4 genes-13-02239-f004:**
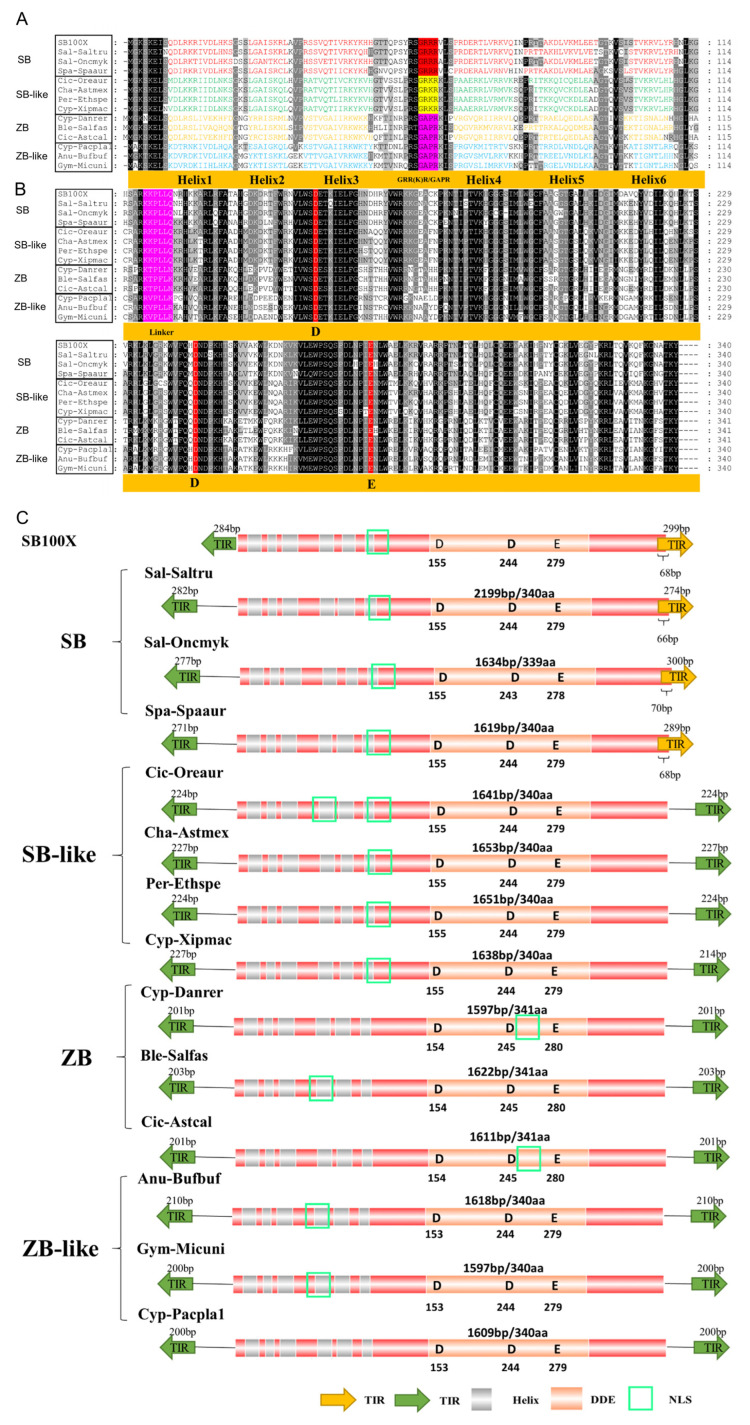
Structural organization of *SB* and *ZB* elements. (**A**) Alignment of the DBD domains of *SB* and *ZB* transposases. Representative species containing complete transposons were selected for each cluster. *Sal-Saltru* (*Salmoniformes*, *Salmo trutta*), *Sal-Oncmyk* (*Salmoniformes*, *Oncorhynchus mykiss*), *Spa-Spaaur* (*Spariformes, Sparus aurata*), *Cic-Oreaur* (*Cichliformes, Oreochromis aureus*), *Cha-Astmex* (*Characiformes, Astyanax mexicanus*), *Per-Ethspe* (*Perciformes, Etheostoma spectabile*), *Cyp-Xipmac* (C*yprinodontiformes*, *Xiphophorus maculatus*), *Cyp-Danrer* (*Cypriniformes*, *Danio rerio*), *Ble-Salfas* (*Blenniiformes, Salarias fasciatus*), *Cic-Astcal* (*Cichliformes, Astatotilapia calliptera*), *Cyp-Pacpla1* (*Cyprinodontiformes, Pachypanchax playfairii*), *Anu-Bufbuf* (*Anura, Bufo bufo*), and *Gym-Micuni* (*Gymnophiona, Microcaecilia unicolor*). Letters in different colors represent HTH patterns, and rectangles in different colors represent GRPR of different transposons. (**B**) Alignment of the DDE domains of *SB* and *ZB* transposases. The red rectangles represent the DDE domain, the pink rectangles represent linkers. (**C**) Structural organization of *SB* and *ZB* elements. Orange arrows and green arrows represent TIRs, and different colors represent different TIRs. The grey light column represents HTH motifs, the yellow circles represent NLS, the orange light column represents catalytic domains, and the red-light column represents transposons.

**Figure 5 genes-13-02239-f005:**
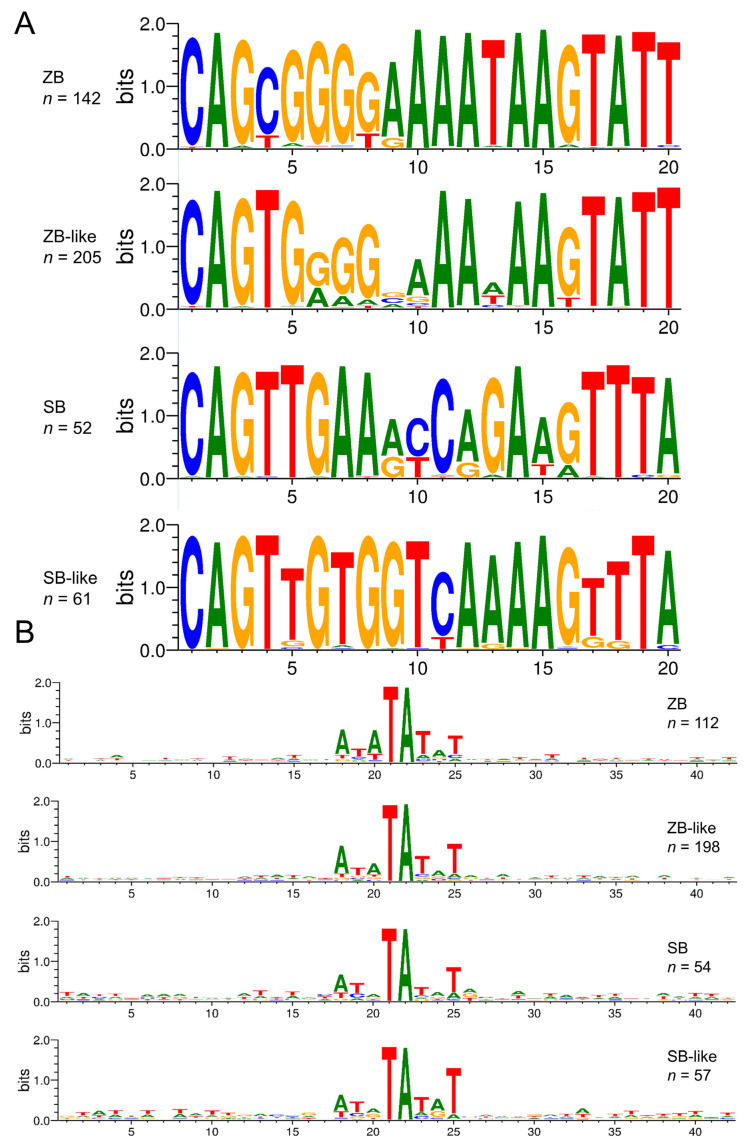
The alignment logo for *SB* and *ZB* elements. (**A**) The Weblogo server (http://weblogo.threeplusone.com/create.cgi, accessed on 16 March 2022) was used to create the logo representation of the first 20 bp of the TIR 5′ sequences. (**B**) Integration sites of four transposons. The sequence logos show the majority regular consensus sequence of the genomic insertion site in a 40 bp window centered on the target TA dinucleotide.

**Figure 6 genes-13-02239-f006:**
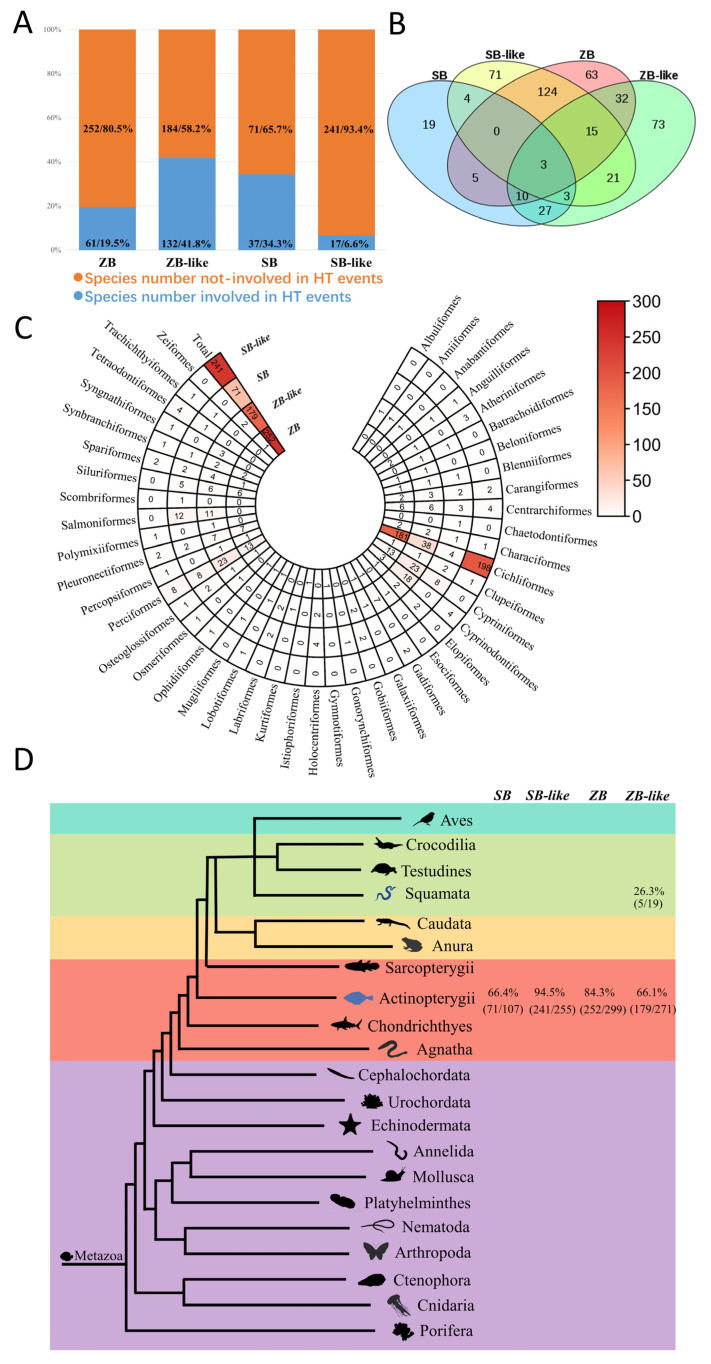
Taxonomic distribution and HT events of *ZB* and *SB* elements. This refers to the HT events jointly supported by two cytoplasmic ribosomal proteins (L3 and L4) in eukaryotic species (detailed information of species was listed in Appendix A). (**A**) The number and proportion of species involved in HT events for the four transposons. (**B**) Venn of species number involved in HT events of *ZB*, *ZB*-like, *SB*, and *SB*-like. (**C**) Species number involved in HT events in Actinopterygii. (**D**) Species number involved in HT events in Metazoa.

**Table 1 genes-13-02239-t001:** Structural organization of *ZB* and *SB* transposons.

Motif	Distribution	Species Number	Length of Full Transposon (bp) ^a^	Length of Intact Transposon (bp) ^b^	Transposase Length (aa) (>300)	TIR Length (bp)	TSD
*ZB*	Anura	9	/	/	/	/	/
	Actinopterygii	299	1448–2940	1513–1687	302–411	113–205	TA
	Mollusca	1	1645	/	/	/	TA
	Arthropoda	4	1601–1622	1601–1622	324–366	198–209	TA
*ZB*-like	Aves	2	/	/	/	/	/
	Squamata	19	1496–2208	1496–2208	338–366	126–214	TA
	Anura	17	1596–1618	1596–1618	340–343	45–211	TA
	Sarcopterygii	1	1597	1597	340	200	TA
	Actinopterygii	271	1331–2063	1391–2063	304–404	53–231	TA
	Chondrichthyes	1	1612	1612	340	201–203	TA
	Agnatha	5	1606–2172	1606–2172	339–340	183–207	TA
*SB*	Actinopterygii	107	858–2979	1332–2979	307–360	44–415	TA
	Mollusca	1	1597	1597	356	267–275	TA
*SB*-like	Aves	1	/	/	/	/	/
	Actinopterygii	255	1571–2111	1621–1753	306–359	105–228	TA
	Echinodermata	1	1816	1816	340	27	TA
	Cnidaria	1	1663	/	/	/	TA

^a^ Full transposon: Transposons flanked by detectable TSDs (target site duplication) and TIRs. ^b^ Intact transposon: Transposons flanked by detectable TSDs and TIRs and encoded ≥ 300 aa transposases.

**Table 2 genes-13-02239-t002:** Full and intact copy numbers of *ZB* and *SB* transposons in genomes.

Family	Distribution	Genome Number				Average ^a^	Genome Number				Average ^b^
		1–9 Full Copy	10–99 Full Copies	≥100 Full Copies	Total		1–9 Intact Copy	10–99 Intact Copies	≥100 Intact Copies	Total	
*ZB*	Actinopterygii	18	108	12	138	44 ± 6.03	46	56	0	102	13 ± 1.15
	Arthropoda	0	1	2	3	438 ± 240.06	2	1	0	3	7 ± 3.21
	Mollusca	0	1	0	1	/	0	0	0	0	/
	Total	18	110	14	142	/	48	57	0	105	-
*ZB*-like	Actinopterygii	18	103	54	175	190 ± 38.21	95	41	19	155	144 ± 41.85
	Agnatha	0	0	4	4	859 ± 389.09	1	0	3	4	332 ± 210.98
	Anura	0	5	4	9	180 ± 89.24	1	5	3	9	560 ± 332.42
	Chondrichthyes	0	0	1	1	/	0	1	0	1	/
	Sarcopterygii	0	0	1	1	/	0	0	1	1	/
	Squamata	0	9	6	15	190 ± 50.32	9	4	0	13	6 ± 1.73
	Total	18	117	70	205	/	106	51	26	183	/
*SB*	Actinopterygii	15	25	13	53	329 ± 98.10	20	4	1	25	19 ± 7.91
	Mollusca	1	0	0	1	/	1	0	0	1	/
	Total	16	25	13	54	/	21	4	1	26	/
*SB*-like	Actinopterygii	9	45	7	61	78 ± 24.99	28	3	4	35	37 ± 17.04
	Cnidaria	1	0	0	1	/	0	0	0	0	/
	Echinodermata	0	1	0	1	/	1	0	0	1	/
	Total	10	46	7	63	/	29	3	4	36	/

^a^ Average full copy (transposons flanked by detectable TSDs and TIRs) number of transposons in genomes, expressed as mean ± SEM. ^b^ Average intact copy (transposons flanked by detectable TSDs and TIRs and encoded ≥ 300 aa transposases) number of transposons in genomes^,^ expressed as Mean ± SEM.

## Data Availability

The data presented in this study are available on request from the corresponding author.

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
