# Peer review of "Horizontal Transfer and Evolutionary Profiles of Two Tc1/DD34E Transposons (ZB and SB) in Vertebrates"

_genes, 2022, doi:10.3390/genes13122239_

Round 1

Reviewer 1 Report

All the figures are missing in the manuscript. I could not review adequately the Results and Discussion sections without them. Please, make amends for this, so that I can adequately review this manuscript.

Reviewer 2 Report

In the present manuscript, authors analyzed the sequence, structure organization, host distribution and phylogenetic relationship of the ZB and SB transposons. Analysis showed that the studied transposons can be classified in four distinct clades, that they are predominately eukaryotic, and mainly Actinopterygii transposons. The conserved sequence and structure suggest the four clades displayed a similar evolution profile, and that they diverged recently. Moreover, authors found that ZB and SB transposons are largely obtained by HT events. Although these results are highly novel for the ZB and even the SB transposons, the authors report that similar events occurred in other transposon families, which suggests this is not uncommon for animal’s transposons.

The manuscript is well written and very easy and interesting to read. The introduction section is well explained and has enough information to understand and support the authors claims and proposed objectives. The methodology approaches used are adequate for the objectives proposed by the authors, they are well described and backed by recent publications. The results are consistent, and they relate to previous findings. Unfortunately, the manuscript Figures and Supplementary material is not available for download, or even to see a preview in the web page, so I could not observe the results described in the results section to evaluate them properly. I understand this in an error and that the Figures must be available, but I could not find them, which hampers my evaluation of the results. Having said this, and assuming this was a mistake, the results are well explained in the text and consistent with the proposed objectives and supported by related references.

Besides this, some minor comments are listed below:

Line 120: first mention of “DBD”, the acronym should be explained (it is explained in line 252 but mentioned five times before that).

Lines 215 to 226: “furthermore” is used in three opportunities in the same paragraph.

Line 239: first mention of “TSD”, the acronym is not explained in the text.

Line 240: please change “>=300aa” for “>=300 aa” to match it to the main text.

Line 254: please check the punctuation in the phrase “Figure. 4A and 4B”.

Line 256: two consecutive punctuations “conserved sequence stretch (KKPLLS) in SB100X transposase [36].. The first…”

Lines 341-344: this is repeated in the methods section (lines 139-141), please check if this was intended.

Comment: In lines 320 to 331, authors stated that “SB-like transposons tend to be more active than SB. The species containing intact copies of SB and SB-like species are much less than that of ZB and ZB-like”. Considering that the sequence and structure organization of the named transposons is highly conserved (lines 176 to 188), do the authors have some explanation for the difference in activities? Or something in the DNA sequence of the transposons, or the catalytic site of the enzymes could suggest a possible answer?

Round 2

Reviewer 2 Report

All figures appear to be in low quality, and the supplementary material is missing or not available to download.

Other comments are listed below.

Line 34: sleeping beauty system is not mentioned enough in the text to include it as a keyword.

Line 108: please change 1e-4 to 1e-4. Please correct the subsequent E-values with the same error.

Figure 1: does not seem to be in high quality… the image is blurred. Also, it appears to be duplicated.

Figure 1 Lines 187-188: the outgroup families are the uncolored zone of the tree? Is not explained and the reference families are not named in the tree.

Figure 2: it is confusing that Figure 2 shows results in a scale from 0 to 1, and the main text explains the results in percentage. Please unify the terms or change the Figure 2.

Line 216: “and n represents the number of each type of…” it’s confusing, maybe highlight the n (use cursive or “n”).

Line 252: change “supplementary table S2” to “Supplementary Table S2”. Supplementary Tables are not available, so I am not sure if it corresponds to Table S2 or Table S3, as it is mentioned in the text in line 246.

Line 276: change “supplementary Figure S2” to “Supplementary Figure S2”.

Figure 4: it is unreadable. The letters are too small and the image quality too low.

Lines 368-369: the phrase “plotted using GraphPad Prism 369 v.8.0.1.244” should be in the materials and methods section.

Figure 6 is unreadable and low quality. Yellow letters in Figure 6A cannot be seen, and the letters in Figure6B and 6C are too small.

Note: I thank the authors for their response to my question in relation to the difference in activities of the transposons. However, it seems this work would of benefit of some experimental validation or exploration of this results. I know this is not the time to ask for this, but maybe the authors should comment some of this response in the discussion section (“The intact copy number of a transposon and sequence identity in a given genome is a KEY factor to judge the activity in genome, high number of intact copy means the transposon have obtained a substantial amplification, and may be still active and can jump and increase copy in genome. However, more accurate prediction of activity can be obtained by combining with more data analysis including protein and DNA sequence identity, DBD and DDE domain, and K divergence”), and include a comment on the relevance of the experimental validation of your results.
